# Healthcare outcomes and special education eligibility in children with congenital CMV

Ryan Rochat[1]*, Elizabeth Goodman[2], Jerry Miller [1], Wei Wang[2], Gail J. Demmler-Harrison[1]

1 Department of Pediatrics, Texas Children's Hospital, Baylor College of Medicine, Houston, Texas, United States of America, 2 Center for Observational and Real World Evidence, Merck & Co., Inc., Rahway, NJ, United States of America

* rochat@bcm.edu

## Abstract

### Objectives

Congenital cytomegalovirus disease (cCMV) can have significant sensory and neurodevelopmental sequelae throughout childhood. Many of these sequalae are consistent with special education eligibility, but the special education needs of affected children have not been systematically studied.

### Methods

Retrospective chart reviews from two cohorts of cCMV children receiving care in a large tertiary care children's hospital were included in this study: a historical research cohort (N = 186, 41% symptomatic at birth) and a contemporary clinical cohort of cCMV patients (N = 112, 68% symptomatic at birth). SNOMED-CT was used to identify ICD-10 codes describing special education-qualifying health outcomes fitting the Individuals with Disabilities Education Act (IDEA) criteria. ICD-10 codes were retrospectively applied to the historical research cohort through manual chart review, whereas the ICD-10 billing codes were extracted from the electronic medical record for all encounters in the contemporary cohort.

### Results

Of the 56 unique special education-qualifying ICD-10 codes we identified as pertinent to the IDEA, at least one was noted in 39% of those asymptomatic at birth (AcCMV) as compared to 94% of the patients symptomatic at birth (ScCMV). In the contemporary clinical cohort, at least one of these codes was noted in 67% of the AcCMV patients as compared to 95% for ScCMV patients. 61% of patients in the historical and 86% in the contemporary cohort had at least one special education-qualifying ICD-10 code. Developmental, mobility, physical therapy and hearing-related health outcomes were common in both ScCMV and AcCMV patients.

### Conclusions

Health outcomes qualifying for special educational services occur commonly in children with cCMV, including those who are classified as asymptomatic at birth. The emergence of

**Data Availability Statement:** Data cannot be shared publicly because of the vulnerable nature of the patients included in this study (children, cognitive impairment), sharing of a full de-identified data set for the purposes of publication is

limited by the Institutional Review Board of Baylor College of Medicine and Affiliated Institutions (Protocol: H-46267) which states "when analyzed and presented or published, will be presented in aggregate data only". As such, data within this manuscript can be made available upon request, and the decision to make available is not restricted by the funder/sponsor. Data are available from the Baylor College of Medicine Institutional Review Board of Baylor College of Medicine and Affiliated Institutions for researchers who meet the criteria for access to confidential data through the following contact information: irb@bcm.edu, phone 713-798-6970. Requests for data should be made to either the corresponding author or IRB (as above).

**Funding:** Drs. Goodman and Wang are employees of Merck Sharp & Dohme LLC, a subsidiary of Merck & Co., Inc., Rahway, NJ, USA and helped conceptualize the study, provided guidance on analytic approach, reviewed and interpreted the results, and edited and reviewed the manuscript. The funder provided support in the form of salaries for all authors (RHR, EG, WW, JM, GJD) but did not have any additional role in the study design, data collection and analysis, decision to publish, or preparation of the manuscript. The specific roles of these authors are articulated in the 'author contributions' section.

**Competing interests:** Drs. Rochat, Miller and Demmler-Harrison received research support from Merck Sharp & Dohme LLC, a subsidiary of Merck & Co., Inc., Rahway, NJ, USA. Drs. Miller and Demmler-Harrison have also received research support from Microgen Laboratories, but this funder was not a source of funding for this study. There are no patents, products in development or marketed products associated with this research. The role of the funder does not alter our adherence to PLOS ONE policies on sharing data and materials as this is prescribed within and by the Institutional Review Board of Baylor College of Medicine and Affiliated Institutions: (H-46267) Health care costs associated with congenital cytomegalovirus (CMV) infection and disease.

**Abbreviations:** CMV, cytomegalovirus; cCMV, congenital cytomegalovirus; AcCMV, asymptomatic congenital cytomegalovirus; ScCMV, symptomatic congenital cytomegalovirus; EMR, electronic medical record; ICD-10, The International Classification of Disease, 10th Revision; SNOMED®, Systematized Nomenclature of Medicine; SNOMED-CT®, Systematized Nomenclature of Medicine Clinical Terms; IDEA, Individuals with Disabilities Education Act.

qualifying conditions beyond the neonatal period among these children suggests that continued surveillance of this vulnerable population throughout early childhood may facilitate the timely identification of health outcomes requiring special educational services.

## Introduction

Cytomegalovirus (CMV) is a leading cause of congenital infection affecting an estimated 0.5 to 1% of live births in the US [1]. Congenital CMV (cCMV) causes sequelae in hearing, vision, growth and development, and is the major non-genetic cause of sensorineural hearing loss. The majority of children with cCMV do not have signs and symptoms at birth and are classified as having asymptomatic (AcCMV) infection, while ~15% are classified as symptomatic (ScCMV) [2]. Children born with ScCMV frequently develop significant neurodevelopmental or sensory sequelae throughout childhood [3–9], and up to 25% of children born with AcCMV may experience progressive sensorineural hearing loss and associated speech and language delays [3].

The phenotypic sensory and neurodevelopmental disabilities associated with cCMV are notable in that many would confer eligibility for special education services under the Individuals with Disabilities Education Act (IDEA). This statute governs how states and public agencies provide early intervention, special education, and related services to infants, toddlers, children, and youth with: "autism, deaf-blindness, deafness, emotional disturbance, hearing impairment, intellectual disability, multiple (concomitant) disabilities, orthopedic impairment, "other" health impairment (covering conditions adversely affecting educational performance), specific learning disability, speech or language impairment, traumatic brain injury, and visual impairment including blindness."[10].

Understanding patterns of both health and educational resource needs is integral in raising awareness for and supporting the complex needs of children with cCMV. To our knowledge, the relationship between cCMV-related health outcomes and resource utilization and special education service eligibility has not been explored. cCMV outcomes studies have focused on a very limited number of health outcomes [11], typically with short follow-up after birth [11–15]. Therefore, emergent sequelae, including developmental disorders and mobility impairments which could relate to special education eligibility, have not been fully described. Furthermore, because medical claims databases, a common source for health outcome studies, have inconsistent documentation of cCMV diagnoses, especially for AcCMV, using such data to study cCMV outcomes and associated healthcare utilization is challenging [12–14, 16–19].

To address these knowledge gaps and data challenges, we examined health outcomes which could confer eligibility for special education accommodation, assistance or placement in two cohorts of children born with cCMV: a historical research cohort and a contemporary clinical cohort. Both cohorts were evaluated at the same tertiary care children's hospital and each provides a unique frame of reference for addressing this question. The historical research cohort enrolled newborns with cCMV between 1982–2005, prior to the introduction of an electronic medical record (EMR). The contemporary clinical cohort includes children born between 2008 (when the hospital introduced its EMR) and 2018. In the historical research cohort, assessments were determined by that study's comprehensive research protocol and were conducted for all participants. Conversely, the contemporary clinical cohort captured all diagnoses, developmental assessments, and associated healthcare utilization during all provider encounters across the study time period. It should be noted that the addition of a

contemporary clinical cohort allows for a real-world approach to disease state classification as the diagnosis and billing codes were derived in aggregate across all providers in our vast multi-center healthcare system and not dependent upon the clinical acumen of single provider. Effectively, the historical cohort provides assessments from a regimented study protocol, while the contemporary cohort relies on assessments from a multitude of providers to determine those patients who may be eligible for accommodations. To that end, we point out that these results should not be used to compare across these cohorts, nor to the general population. The unique and complementary strengths of these two cohorts studied together allow us to over-come the challenges of using either alone to study this question in children with cCMV and allows us to detect a similar signal that may be present for these children using two distinct approaches.

## Methods

### Study populations

This is a retrospective study of two distinct cCMV cohorts. The historical research cohort, was derived from the Houston Longitudinal Congenital CMV Study, which enrolled newborns with cCMV from 1982–2005 [4]. Houston Longitudinal Congenital CMV Study subjects were identified at birth from a large maternity hospital through universal newborn screening and from outside referrals from 1982–2005 [4, 20, 21]. All patients had CMV infection confirmed by CMV culture of urine samples collected within 3 weeks of life. Subjects who had at least one of the following CMV-related signs at birth were classified as ScCMV: purpura/petechiae, jaundice, hepatosplenomegaly, microcephaly, unexplained neurological abnormality, elevated liver enzymes (alanine aminotransferase > 100 IU), hyperbilirubinema (total bilirubin > 3 mg dl$^{-1}$), hemolytic anemia or thrombocytopenia (platelet count < 75,000 mm$^3$) [4, 6, 7, 22, 23]. Patients with CMV infection, but none of these CMV-related signs at birth, were classified as AcCMV. Of these historical cohort study participants, 41.4% (n = 77) were ScCMV and 58.6% (n = 109) AcCMV. Subjects were followed prospectively with age-appropriate study visits including assessments in growth, hearing, vision, IQ, academic performance, and development [3–5, 8, 9, 20]. The historical cohort comprised 2,377 person-years of follow-up on its 186 patients up to age 18, with a mean of 12.8 years of follow-up.

The contemporary clinical cohort is comprised of patients with cCMV who were born between 2008–2018 and followed through 2020. cCMV was confirmed by detection of CMV in urine, saliva or blood—including newborn dried blood spot—by CMV culture or CMV DNA PCR. Using SlicerDicer$^{TM}$, a reporting tool in our hospital's Epic® EMR, we identified 112 patients who met these criteria, and were seen at least once by the cCMV Clinic Director (GDH), to be specific in our assignment of AcCMV/ScCMV. As with the historical research cohort, patients with CMV infection, but none of the CMV-related signs at birth, were classi-fied as AcCMV Patients in the contemporary clinical cohort were diagnosed with cCMV dur-ing real-world medical evaluations, and in contrast to the historical research cohort, were not part of prospective research study. As there was no universal screening protocol to identify the patients in the contemporary cohort, inclusion in this cohort was on the basis of referral to our clinic from any birthing hospital or provider in the greater Houston area. As with the historical cohort, patients in the contemporary cohort who had at least one CMV-related sign at birth were classified as ScCMV (67.9%, N = 76). It should be noted that the diagnoses and healthcare utilization in the contemporary clinical cohort were not assigned by the authors, instead they reflect actual healthcare needs, encounters and utilization as captured across our healthcare system. The contemporary cohort comprised 771 person-years of follow up on its 112 patients, with a mean of 6.9 years of follow-up per patient, spanning nearly 12,000 provider encounters.

This study as well as data collection and analysis for this project was approved by the Institutional Review Board of Baylor College of Medicine and Affiliated Institutions approved (Protocol: H-46267) "Health care costs associated with congenital cytomegalovirus (CMV) infection and disease." As this was a retrospective chart review, consent was waived by the Baylor College of Medicine and Affiliated Institutions Institutional Review Board.

## Data acquisition

Records from the historical cohort consisted of paper-based study charts and previously curated databases [4, 20, 21]. These records, which did not include diagnosis codes, were manually reviewed by two of the investigators (JM, GDH) and applicable codes were retroactively assigned to health outcomes which had clear documentation paper-based study charts, consistent with International Classification of Disease, 10th Revision (ICD-10) codes (S1 Table). As such, all codes for the historical cohort were abstracted from original study documentation as ICD-10 codes. Complex or ambiguous diagnoses were further adjudicated by the two investigators using chart reviews, study databases and knowledge of the patients.

For the contemporary cohort, the EMR data was accessed through the Epic® data warehouse. Patient encounters and associated billing diagnoses from 2008–2020 were extracted from the EMR, which was followed by manual adjudication of cCMV classification (RHR, GDH). In accordance with regulations from the Centers for Medicare & Medicaid Services, any ICD-9 codes within our EHR had been migrated to ICD-10 and so a *post hoc* reassignment of diagnostic codes was unnecessary. All available records up to the age of 18 years for both cohorts were included in the present study.

## Linking ICD-10 coding to IDEA Act eligibility

The International Classification of Disease, 10th Revision (ICD-10), is the global health information standard for documenting medical diagnoses. Its codes are used for claims processing, to manage healthcare, monitor outcomes, and allocate health resources. To develop linkages between ICD-10 coding and eligibility for special education services under the IDEA Act, we used the Systematic Nomenclature of Medicine Clinical Terms system (SNOMED-CT®). SNOMED-CT® is often integrated with the EMR to assist providers in mapping problems to appropriate ICD-10 diagnosis codes. To harmonize the health outcomes and diagnostic codes for these two cohorts with the broad categories listed above in the IDEA Act, SNOMED-CT® was used to define seven specific systems-based health outcome categories of ICD-10 codes: hearing problems, hearing devices, central nervous system (CNS) problems, vision problems, neuro-muscular problems, developmental delays and disabilities, and daily living, functioning, and assistive aids (S1 Table). These seven categories serve as a bridge between the language of the IDEA Act and the medically precise, granular ICD-10 codes abstracted from the study charts and medical records.

## Analyses

Basic descriptive statistics were derived using SPSS (Version 27, IBM, Inc., Armonk, NY) for the historical cohort and SAS Software V9.4 (SAS Institute Inc., Cary, NC, USA) for the contemporary cohort. For both cohorts, the counts, and percentages of (special education-eligible and total) subjects having the health outcomes of interest were tabulated and summed as appropriate, and the mean, median and range of the number of health outcomes reported. Differences in proportions across categorical variables were assessed using chi-square tests. For the contemporary cohort, provider/encounter specific billing data enabled us to calculate the

age at which a diagnosis was first noted in our EMR, which we restricted to only those patients who initiated care within our healthcare system by 6 months of age.

The two cohorts described above are derived from two different populations of children with cCMV, one a formalized prospective research cohort, and the other a real-world clinical cohort captured by virtue of an informatic analysis of our extensive EMR documentation. The differences in these two datasets (e.g. time period, formal evaluation, referral bias, etc.) make direct comparison of health outcomes and special education needs between these two cohorts unsuitable. Rather, we provide descriptive analyses of each cohort to corroborate and validate our findings of cCMV-related special education-eligible health outcomes across a research and real-world cohort.

## Results

Demographic and birth characteristics are presented in Table 1. Patients were primarily non-Hispanic white. There was greater non-white representation in the contemporary compared to the historical cohort [Hispanic (40% vs 18%, $p < .001$) and African American (30% vs. 15%, $p < .003$)], which may reflect the different referral base, changing demographics of the metropolitan area, and willingness to participate in research between the two cohort accrual periods. The lower mean gestational age and birthweight of the contemporary compared to the historical cohort are attributable to exclusion criteria (e.g. excluding extreme prematurity) in the Houston Longitudinal Congenital CMV Study.

**Table 1. Demographic data for newborns in the historical and contemporary cCMV cohorts.**

|  | Historical Research Cohort (N = 186) | Contemporary Clinical Cohort (N = 112) |
|---|---|---|
| Male, n(%) | 90 (48) | 59 (53) |
| Race/Ethnicity** |  |  |
| Non-Hispanic White | 128 (69) | 48 (42) |
| Hispanic | 29 (16) | 26 (23) |
| Non-Hispanic Black | 27 (15) | 32 (28) |
| Asian | 2 (1) | 5 (4) |
| Other | 0 (0) | 1 (1) |
| Mother's Age (y) |  |  |
| Mean (SD) | 26.6 (6.3) | 25.3 (5.8) |
| Median | 26 | 24.7 |
| Range | 13.0–46.0 | 16.0–37.4 |
| Estimated Gestational Age at Birth (weeks) |  |  |
| Mean (SD) | 38.0 (2.2) | 36.2 (4.3) *** |
| Median | 38.3 | 37 |
| Range | 32.0–41.9 | 24–41 |
| Birthweight (g) |  |  |
| Mean (SD) | 2887 (781) | 2391 (703) *** |
| Median | 2865 | 2480 |
| Range | 1085–4975 | 600–3700 |
| Symptomatic at birth | 77 (41) | 76 (68) |

** $0.001 < p < 0.01$.

*** $p < .0001$.

## Historical research cohort

Table 2 presents health outcomes in the historical research cohort which may qualify for special educational placement, assistance or accommodation under the IDEA Act, stratified by the seven specific systems-based health outcome categories. Overall, 61% of patients in the historical cohort had health outcomes which could confer special education eligibility. The most prevalent health outcome category present that would warrant special education was Developmental Disabilities and Delays, with 56% having health outcomes in this category. Hearing problems were common, with 29% having moderate or worse hearing loss; and 28% having hearing devices. Health outcomes in the category of daily living, functioning and assistive aids affected 27% of subjects in the historical research cohort.

Health outcomes involving neuromuscular problems, developmental delays and disabilities, and daily living, functioning and assistive aids were commonly found among AcCMV subjects in the historical research cohort. In this cohort AcCMV patients were noted to have health outcomes in neuromuscular (6%), developmental (33%), and CNS problems (22%). Finally, a substantial proportion of AcCMV subjects in historical research cohort (17%) had hearing deficits that would qualify for special education eligibility.

## Contemporary clinical cohort

Table 3 presents the unique health outcomes attributed to children in the contemporary clinical cohort across all healthcare encounters in our medical system. Overall, 86% of these children had at least one health outcome documented in the EMR which could confer special education eligibility under the IDEA Act. As with the historical research cohort, deficits were most often noted in the category of Developmental Disabilities and Delays, with 71% of all children affected. Hearing problems were common, with 49% having moderate or worse hearing loss; and 32% having hearing devices. Health outcomes in the category of daily living, functioning and assistive aids affected 25% of children in the contemporary clinical cohort.

Given the granularity of the data in the EMR, we are able to determine a timeline for outcomes for the contemporary clinical cohort. Of the seven specific systems-based health outcome categories identified, deficits in hearing were noted within the first year of life (mean of 0.84 years), followed by vision (1.2 years) and neuro-muscular problems (1.8 years). All children, who were noted to have IDEA Act qualifying outcomes, presented within the first 3 years of life with deficits in one or more of these seven systems, highlighting the value of longitudinal assessment in these patients.

As with the historical research cohort, detrimental health outcomes were commonly found among children classified as having AcCMV at birth. AcCMV patients in the contemporary clinical cohort were noted to have eligible health outcomes in the following systems: neuro-muscular (39%), developmental (56%), vision (6%), and CNS (22%). Lastly, a substantial proportion of AcCMV subjects (31%) had hearing deficits that would qualify for special education eligibility.

## Cohort comparison

Overall, in the historical cohort, 94% of ScCMV patients and 39% of the AcCMV patients had conditions that would be qualifying under the IDEA Act as eligible for special education, while in the contemporary cohort, 95% by ScCMV patients and 67% of AcCMV patients had qualifying conditions. While subjects with problems in daily living/functioning/assistive aids were more prevalent among ScCMV patients in the historical cohort than in the contemporary cohort, this likely represents a reflection of the prospective study design of the historical cohort with corresponding age-appropriate study visits, assessments, and longer follow up time.

**Table 2. Health outcomes potentially qualifying for special educational placement, assistance or accommodation in historical research cohort (N = 186).**

| Health Outcome Category and Related ICD-10 Descriptions | IDEA Act Qualifying Condition* | ICD-10 Code | Number (%) of AcCMV Subjects (n = 109) | Number (%) of ScCMV Subjects (n = 77) | Total Affected Subjects: n (% of cohort) |
|---|---|---|---|---|---|
| Hearing Problems | | | 18 (17) | 36 (47) | 54 (29) |
| SNHL Moderate or Worse ASHL Level** | HI | H90.5 | 18 (17) | 36 (47) | 54 (29) |
| Auditory Processing Disorder | HI | H93.25 | 0 (0) | 1 (1) | 1 (1) |
| SNHL Preferential Classroom Seating | HI | (no code) | 10 (9) | 0 (0) | 10 (5) |
| SNHL Special Education | HI | (no code) | 2 (2) | 0 (0) | 2 (1) |
| Hearing Devices | | | 11 (10) | 41 (53) | 52 (28) |
| Cochlear Implant | HI | Z96.21 | 1 (1) | 15 (20) | 16 (9) |
| Hearing Aid | HI | Z97.4 | 9 (8) | 37 (48) | 46 (25) |
| SNHL Assistive Listening Device | HI | F06ZBLZ | 8 (7) | 0 (0) | 8 (4) |
| CNS Problems | | | 0 (0) | 29 (39) | 29 (16) |
| Cerebral Palsy: Spastic Quadriplegic Type | MCD, OI | G80.0 | 0 (0) | 18 (25) | 18 (10) |
| Cerebral Palsy: Spastic Diplegic Type | MCD, OI | G80.1 | 0 (0) | 1 (1) | 1 (1) |
| Cerebral Palsy: Athetoid Type | MCD, OI | G80.3 | 0 (0) | 1 (1) | 1 (1) |
| Cerebral Palsy: Ataxic Type | MCD, OI | G80.4 | 0 (0) | 1 (1) | 1 (1) |
| Cerebral Palsy: Unspecified Type | MCD, OI | G80.9 | 0 (0) | 2 (3) | 2 (1) |
| Vision Problems | | | 0 (0) | 29 (40)[G] | 29 (16) |
| Vision—Retinal Detachment | VI | H33.20 | 0 (0) | 1 (1) | 1 (1) |
| Vision—Optic Atrophy | VI | H47.20 | 0 (0) | 7 (9) | 7 (4) |
| Vision—Cortical Visual Impairment | VI | H47.619 | 0 (0) | 11 (14) | 11 (6) |
| Vision—Severe Acuity at Most Recent Exam | VI | H53.8 | 0 (0) | 10 (14) | 10 (5) |
| Neuro-muscular Problems | | | 7 (6) | 34 (44) | 41 (22) |
| Motor Function Disorder | OI | F82 | 5 (5) | 15 (19) | 20 (11) |
| Tendon Contracture | OI | M62.4 | 0 (0) | 19 (25) | 19 (10) |
| Muscular Hypertonia | OI | P94.1 | 0 (0) | 3 (4) | 3 (2) |
| Unable to Release Grip | OI | R29.898 | 0 (0) | 6 (8) | 6 (3) |
| Muscular Hypotonia | OI | R29.898 | 2 (2) | 1 (1) | 3 (2) |
| Delay in Physiological Development | MCD, OTHER | R62.50 | 0 (0) | 14 (18) | 14 (8) |
| Developmental Delays and Disabilities | | | 36 (33) | 68 (88) | 104 (56) |
| Mild Intellectual Disability IQ50-70 | ID | F70 | 1 (1) | 14 (18) | 15 (8) |
| Moderate Intellectual Disability IQ35-55 | ID | F71 | 0 (0) | 4 (5) | 4 (2) |
| Severe Intellectual Disability IQ20-40 | ID | F72 | 0 (0) | 5 (6) | 5 (3) |
| Profound Intellectual Disability IQ25 | ID | F73 | 0 (0) | 15 (19) | 15 (8) |
| Intellectual Disability Unspecified | ID | F79 | 0 (0) | 2 (3) | 2 (1) |
| Other Intellectual Disability | ID | F78 | 0 (0) | 3 (4) | 3 (2) |
| Developmental Articulation Disorder | SLI, SLD | F80.0 | 0 (0) | 14 (18) | 14 (8) |
| Expressive Language Disorder | SLI | F80.1 | 4 (4) | 56 (73) | 60 (32) |
| Mixed Receptive-Expressive Language Disorder | SLI | F80.2 | 3 (3) | 26 (34) | 29 (16) |
| Speech and Language Developmental Delay Due to Hearing Loss | SLI, HI | F80.4 | 8 (7) | 21 (27) | 29 (16) |
| Speech Language Disorders of Development | SLI | F80.8 | 0 (0) | 6 (8) | 6 (3) |
| Developmental Disorder of Speech and Language | SLI | F80.9 | 0 (0) | 32 (42) | 32 (17) |
| Scholastic Learning Disorder | SLD | F81.9 | 3 (3) | 15 (19) | 18 (10) |
| Autism | AUT | F84.0 | 0 (0) | 12 (16) | 12 (6) |

*(Continued)*

**Table 2.** (Continued)

| Health Outcome Category and Related ICD-10 Descriptions | IDEA Act Qualifying Condition* | ICD-10 Code | Number (%) of AcCMV Subjects (n = 109) | Number (%) of ScCMV Subjects (n = 77) | Total Affected Subjects: n (% of cohort) |
|---|---|---|---|---|---|
| Other Pervasive Developmental Disorders | MCD, OTHER | F84.8 | 0 (0) | 8 (10) | 8 (4) |
| Global Developmental Delay | MCD, OTHER | F88 | 0 (0) | 23 (30) | 23 (12) |
| Psychological Developmental Disorder | EM | F89 | 0 (0) | 8 (10) | 8 (4) |
| Speech and Language Therapy | SLI | F06ZBZZZ | 7 (6) | 8 (10) | 15 (8) [8] |
| Borderline Intellectual Disability IQ70-85 | ID | R41.83 | 0 (0) | 3 (4) | 3 (2) |
| Delayed Milestone | OTHER | R62.0 | 2 (2) | 47 (61) | 49 (26) |
| Daily Living, Functioning, and Assistive Aids | | | 7 (16) | 43 (56)[A] | 50 (27) |
| Difficulty Walking | OI | R26.2 | 0 (0) | 21 (27) | 21 (11) |
| Balance Problems | OI | R27.8 | 1 (1) | 4 (5) | 5 (3) |
| Reduced Mobility | OI | Z74.0 | 0 (0) | 21 (27) | 21 (11) |
| Needs Walking Aid | OI | Z74.09 | 0 (0) | 19 (25) | 19 (10) |
| Requires Assistance with All Daily Activities | OI, MCD | Z74.1 | 0 (0) | 24 (31) | 24 (13) |
| Occupational Therapy | MCD, OI, OTHER | Z76.89 | 0 (0) | 35 (45) | 35 (19) |
| Gastrostomy Present ("G-tube") | OTHER | Z93.1 | 0 (0) | 15 (19) | 15 (8) |
| Dependence on Wheelchair | OI | Z99.3 | 0 (0) | 16 (21) | 16 (9) |
| Physical Therapy | MCD, OI, OTHER | F07M3ZZ | 0 (0) | 32 (42) | 32 (17) |
| Dependence on Orthoses | OI | Z99.89 | 0 (0) | 20 (26) | 20 (11) |
| Vision Therapy | VI | (no code) | 0 (0) | 2 (3) | 2 (1) |

* IDEA Act Qualifying Conditions Codes: A Autism; HI Hearing Impairment; EM Emotional Disturbance; ID Intellectual Disability; MCD Multiple (Concomitant) Disabilities; OI Orthopedic Impairment; OTHER Other Health Impairment (adversely affecting educational performance); SLD Specific Learning Disability; SLI Speech or Language Impairment; VI Visual Impairment. These represent the most plausible major qualifying conditions, although most subjects could qualify with MCD (Multiple, Concomitant Disabilities).

**ASHL American Speech-Language Hearing Association (28).

Table 4 presents summary statistics for health outcomes potentially qualifying for special educational placement, assistance or accommodation. Nearly all ScCMV subjects in both cohorts had at least one eligible outcome, while 39% of the AcCMV subjects in the historical cohort and 67% of the AcCMV patients in in the contemporary cohort were similarly affected. When looking at children with three or more eligible outcomes, a majority of children with ScCMV were noted to satisfy these criteria, as were a substantial number with AcCMV.

The provider-specific healthcare utilization of children in the contemporary clinical cohort is shown, in part, in Table 5. These records reflect billed encounters within the noted specialty and are a direct reflection of clinical care provided. An in-depth analysis of this data is beyond the scope of the current paper, however these encounters serve to reenforce the results in Tables 3 and 4, and confirm that these sequalae translate to real-world delivery of healthcare services. Notably, despite being a congenital infection, only 5% of all provider encounters for this cohort were in Infectious Diseases.

## Discussion

This study demonstrates that health outcomes qualifying for special educational assistance under the IDEA Act and associated special education guidelines are prevalent for children with cCMV, including those asymptomatic (i.e. having no CMV-related signs) at birth, and highlights the previously undocumented educational needs of these children. Our utilization of both historical research and contemporary clinical cCMV cohorts and use of

**Table 3. Health outcomes potentially qualifying for special educational placement, assistance or accommodation in contemporary clinical cohort (N = 112).**

| Health Outcome Category and Related ICD-10 codes | IDEA Act Qualifying Condition | ICD-10 Code | Number (%) of AcCMV Subjects (n = 36) | Number (%) of ScCMV Subjects (n = 76) | Total Affected Subjects n (% of cohort) | Age First Noted (years): Mean (SD) [Median] |
|---|---|---|---|---|---|---|
| Hearing Problems | | | 11 (31) | 44 (58) | 55 (49) | 0.84 (1.0) [0.4] |
| SNHL Moderate or Worse ASHL Level* | HI | H90.5 | 11 (31) | 44 (58) | 55 (49) | 0.84 (1.0) [0.39] |
| Auditory Processing Disorder | HI | H93.25 | 0 (0) | 1 (1) | 1 (1) | 1.04 (N/A) [1.04] |
| Hearing Devices | | | 8 (22) | 28 (37) | 36 (32) | 2.6 (2.2) [2.0] |
| Cochlear Implant | HI | Z96.21 | 3 (8) | 14 (18) | 17 (16) | 3.2 (2.6) [2.1] |
| Hearing Aid | HI | Z97.4 | 5 (14) | 14 (18) | 19 (17) | 2.3 (1.9) [2.0] |
| CNS Problems | | | 8 (22) | 38 (50) | 46 (41) | 2.7 (2.7) [1.7] |
| Cerebral Palsy: Spastic Quadriplegic Type | MCD, OI | G80.0 | 2 (6) | 10 (13) | 12 (11) | 3.4 (2.6) [2.3] |
| Cerebral Palsy: Spastic Diplegic Type | MCD, OI | G80.1 | 2 (6) | 4 (5) | 6 (5) | 6.7 (4.3) (9.1) |
| Cerebral Palsy: Athetoid Type | MCD, OI | G80.3 | 0 (0) | 0 (0) | 0 (0) | N/A |
| Cerebral Palsy: Ataxic Type | MCD, OI | G80.4 | 0 (0) | 0 (0) | 0 (0) | N/A |
| Cerebral Palsy: Unspecified Type | MCD, OI | G80.9 | 7 (19) | 17 (22) | 24 (22) | 2.1 (1.8) [1.5] |
| Vision Problems | | | 2 (6) | 7 (9) | 9 (8) | 1.2 (1.2) [0.7] |
| Vision—Retinal Detachment | VI | H33.20 | 0 (0) | 1 (1) | 1 (1) | 0.3 (N/A) [0.3] |
| Vision—Optic Atrophy | VI | H47.20 | 1 (3) | 2 (3) | 3 (3) | 1.6 (1.2) [1.5] |
| Vision—Cortical Visual Impairment | VI | H47.619 | 0 (0) | 7 (9) | 7 (6) | 1.4 (1.3) [0.5] |
| Vision—Severe Acuity at Most Recent Exam | VI | H53.8 | 1 (3) | 4 (5) | 5 (4) | 1.03 (1.5) [0.3] |
| Neuro-muscular Problems | | | 14 (39) | 56 (74) | 70 (63) | 1.8 (1.9) [1.1] |
| Motor Function Disorder | OI | F82 | 7 (19) | 22 (29) | 29 (26) | 2.1 (1.6) [1.4] |
| Tendon Contracture | OI | M62.4 | 5 (14) | 29 (38) | 34 (30) | 1.01 (1.4) [0.5] |
| Muscular Hypertonia | OI | P94.1 | 1 (3) | 6 (8) | 7 (6) | 1.69 (2.2) [0.6] |
| Unable to Release Grip | OI | R29.898 | 0 (0) | 0 (0) | 0 (0) | NA |
| Muscular Hypotonia | OI | R29.898 | 3 (8) | 18 (24) | 21 (19) | 3.3 (2.7) [2.8] |
| Delay in Physiological Development | MCD, OTHER | R62.50 | 10 (33) | 45 (59) | 55 (49) | 1.1 (1.4) [0.65] |
| Developmental Delays and Disabilities | | | 20 (56) | 60 (79) | 80 (71) | 1.9 (1.6) [1.6] |
| Mild Intellectual Disability IQ50-70 | ID | F70 | 0 (0) | 0 (0) | 0 (0) | N/A |
| Moderate Intellectual Disability IQ35-55 | ID | F71 | 0 (0) | 0 (0) | 0 (0) | N/A |
| Severe Intellectual Disability IQ20-40 | ID | F72 | 0 (0) | 1 (1) | 1 (1) | 2.7 (N/A) [2.7] |
| Profound Intellectual Disability IQ25 | ID | F73 | 0 (0) | 1 (1) | 1 (1) | 5.1 (N/A) [5.1] |
| Intellectual Disability Unspecified | ID | F79 | 2 (6) | 4 (5) | 6 (5) | 5.1 (3.2) [5.6] |
| Other Intellectual Disability | ID | F78 | 0 (0) | 0 (0) | 0 (0) | N/A |
| Developmental Articulation Disorder | SLI, SLD | F80.0 | 2 (6) | 4 (5) | 6 (5) | 4.3 (1.2) [3.9] |
| Expressive Language Disorder | SLI | F80.1 | 8 (22) | 14 (18) | 22 (20) | 1.9 (1.1) [1.6] |
| Mixed Receptive-Expressive Language Disorder | SLI | F80.2 | 5 (14) | 15 (20) | 20 (18) | 2.3 (1.6) [2.0] |
| Speech and Language Developmental Delay Due to Hearing Loss | SLI, HI | F80.4 | 2 (6) | 11 (14) | 13 (12) | 2.7 (2.0) [1.8] |
| Speech Language Disorders of Development | SLI | F80.8 | 7 (19) | 24 (32) | 31 (28) | 1.9 (1.0) [1.9] |
| Developmental Disorder of Speech and Language | SLI | F80.9 | 17 (47) | 34 (45) | 51 (46) | 2.2 (1.6) [1.4] |
| Scholastic Learning Disorder | SLD | F81.9 | 0 (0) | 0 (0) | 0 (0) | N/A |
| Autism | AUT | F84.0 | 4 (11) | 8 (11) | 12 (11) | 2.5 (0.6) [2.3] |

*(Continued)*

Table 3. (Continued)

| Health Outcome Category and Related ICD-10 codes | IDEA Act Qualifying Condition | ICD-10 Code | Number (%) of AcCMV Subjects (n = 36) | Number (%) of ScCMV Subjects (n = 76) | Total Affected Subjects n (% of cohort) | Age First Noted (years): Mean (SD) [Median] |
|---|---|---|---|---|---|---|
| Other Pervasive Developmental Disorders | MCD, OTHER | F84.8 | 0 (0) | 0 (0) | 0 (0) | N/A |
| Global Developmental Delay | MCD, OTHER | F88 | 10 (28) | 25 (33) | 35 (31) | 1.4 (1.5) [1.1] |
| Psychological Developmental Disorder | EM | F89 | 0 (0) | 1 (1) | 1 (1) | 1.5 (N/A) [1.5] |
| Borderline Intellectual Disability IQ70-85 | ID | R41.83 | 0 (0) | 0 (0) | 0 (0) | N/A |
| Delayed Milestone | OTHER | R62.0 | 5 (14) | 30 (39) | 35 (31) | 0.9 (1.1) [0.5] |
| Daily Living, Functioning, and Assistive Aids | | | 5 (14) | 23 (30) | 28 (25) | 2.8 (2.6) [1.7] |
| Difficulty Walking | OI | R26.2 | 0 (0) | 2 (3) | 2 (2) | 3.7 (N/A) 3.7 |
| Balance Problems | OI | R27.8 | 5 (14) | 13 (17) | 18 (16) | 1.1 (1.2) [0.7] |
| Reduced Mobility | OI | Z74.0 | 2 (6) | 8 (11) | 10 (9) | 2.5 (2.0) [1.7] |
| Needs Walking Aid | OI | Z74.09 | 2 (6) | 8 (11) | 10 (9) | 2.5 (2.0) [1.7] |
| Requires Assistance with All Daily Activities | OI, MCD | Z74.1 | 0 (0) | 0 (0) | 0 (0) | N/A |
| Occupational Therapy | MCD, OI, OTHER | Z76.89 | 1 (3) | 1 (1) | 2 (2) | 0.6 (0.4) [0.5] |
| Gastrostomy Present (G-tube) | OTHER | Z93.1 | 0 (0) | 13 (17) | 13 (12) | 2.2 (2.7) [1.4] |
| Dependence on Wheelchair | OI | Z99.3 | 0 (0) | 0 (0) | 0 (0) | 3.7 (3.0) [2.9] |
| Dependence on Orthoses | OI | Z99.89 | 1 (3) | 2 (3) | 3 (3) | 8.5 (3) [5.1] |

* IDEA Act Qualifying Conditions Codes: A Autism; HI Hearing Impairment; EM Emotional Disturbance; ID Intellectual Disability; MCD Multiple (Concomitant) Disabilities; OI Orthopedic Impairment; OTHER Other Health Impairment (adversely affecting educational performance); SLD Specific Learning Disability; SLI Speech or Language Impairment; VI Visual Impairment. These represent the most plausible major qualifying conditions, although most subjects could qualify with MCD (Multiple, Concomitant Disabilities).

**ASHL American Speech-Language Hearing Association (28).

SNOMED-CT®/ICD-10 nomenclature as a "common language" to harmonize across them are novel aspects of this study, which expands knowledge of the burden of cCMV disease. Our approach to relate these two cohorts through SNOMED mapping allowed us to explore nearly 40 years of follow-up and highlight an important, heretofore unrecognized consistent finding across these two cohorts: health outcomes and healthcare resource utilization related to special education eligibility are common in children with cCMV, even those classified as asymptomatic at birth.

The complex nature of many of the identified health outcomes, which often involve neuro-sensory pathologies, suggests they could contribute to learning disabilities and challenges that school districts are obligated to address. The majority of children in both the historical and

Table 4. Summary table for health outcomes potentially qualifying for special educational placement, assistance or accommodation.

| Summary Statistics | Historical Research Cohort | | | Contemporary Clinical Cohort | | |
|---|---|---|---|---|---|---|
| | AcCMV (n = 109) | ScCMV (n = 77) | Total (n = 186) | AcCMV (n = 36) | ScCMV (n = 76) | Total (n = 112) |
| | N (%) | N (%) | N (%) | N (%) | N (%) | N (%) |
| Unique Subjects with One or More Qualifying Outcomes: | 42 (39) | 72 (94) | 114 (61) | 24 (67) | 72 (95) | 96 (86) |
| Unique Subjects with Two or More Qualifying Outcomes: | 22 (20) | 68 (88) | 89 (48) | 16 (44) | 62 (82) | 78 (70) |
| Unique Subjects with Three or More Qualifying Outcomes: | 13 (12) | 63 (82) | 76 (41) | 12 (33) | 52 (68) | 64 (57) |

**Table 5. Listing of the most frequent provider encounters for patients in the contemporary cohort.**

| Provider Encounter | Encounters n = 11,653 (% of provider encounters) |
|---|---|
| Physical Therapy | 2879 (25) |
| Occupational Therapy | 1548 (13) |
| Speech Therapy | 1178 (10) |
| Audiology | 701 (6) |
| Infectious Diseases | 587 (5) |
| Otolaryngology | 293 (3) |
| Ophthalmology | 277 (2) |
| Neurology | 237 (2) |
| Anesthesia | 715 (6) |
| Surgery | 274 (2) |

Listing of the most frequent provider encounters for all 112 patients with cCMV within the contemporary cohort. Surgery includes any operating room encounter by general surgery, orthopedic surgery, neurosurgery, ophthalmology, and otolaryngology and covers all types of surgical procedures.

contemporary cohorts had conditions which would make them eligible for special education services. While the majority of those eligible have ScCMV, two-thirds of AcCMV subjects in the contemporary cohort and more than one-third in the historical cohort had qualifying conditions, often related to sensorineural hearing loss at birth or emergent during childhood. The greater proportion of AcCMV patients in the contemporary cohort with qualifying conditions may reflect the real-world evaluation of these children across numerous providers. Compared to the historical cohort, where evaluations for these patients was limited to a single provider, a real-world aggregation across providers may increase sensitivity to detect emergent sequelae.

Because hearing loss alone is generally not used to classify patients as symptomatic at birth, these finding suggest that providers should remain vigilant for the impact cCMV can have on education-related development well beyond infancy and underscores the need to move past classifications of symptomatic or asymptomatic disease assigned at birth in defining this illness. This is further supported by our data from the contemporary cohort in which we were able to calculate the age at which these diagnosis were first noted, nearly all of which were outside of the neonatal period. Furthermore, the encounter-level data, which captured the full extent of patient interactions within our healthcare system, suggest that it is not just the infectious disease physician who interacts with cCMV- affected children, and highlights the multidisciplinary needs of this patient population, particularly in relation to physical/occupational/speech therapy and a range of developmental disabilities. Ultimately, the healthcare resource utilization pattern we see in the contemporary cohort suggests that the role of the infectious disease specialist fades after the initial diagnostic and initial management phases, while other healthcare services take on a larger role as children develop.

## Limitations

While our use of two distinct cohorts allowed us to document special education-related cCMV outcomes across 4 decades, these findings cannot be generalized to all children with cCMV or be used to derive population-level rates. Our retrospective assignment of codes to subjects in the historical cohort could introduce bias, however this is balanced by the potential for underreporting of relevant health outcomes in the Houston Longitudinal Congenital CMV Study, since special educational assistance was not a focus of that study. Lastly, as our clinical site is a

referral center for children with cCMV, there is potential for referral bias in the contemporary cohort, although we believe this bias is minimal, as we had similar findings for patients in the historical cohort, who were identified through universal screening.

## Conclusions

This study demonstrates the widespread special education needs of two separate cohorts of children with cCMV spanning 40 years. Furthermore, this work highlights the evolving manifestations and ramifications of cCMV as children age into adolescence. The developmental and hearing deficits noted for AcCMV patients suggests that early identification and monitoring of these children may facilitate early entry into needed educational interventions. Health policy makers, researchers, and economists should consider these health outcomes and educational service needs when calculating the impact of cCMV disease. Finally, as more healthcare systems move towards EMRs there is a unique opportunity to study the progression of disease in congenital infections with unprecedented detail by leveraging the power of clinical informatics in these analyses.

## Supporting information

**S1 Table. SNOMED-CT® ICD-10 code groupings.** Listing of ICD-10-CM codes that have been grouped as guided by the SNOMED-CT.
(PDF)

## Acknowledgments

The researchers wish to acknowledge and thank the patients and families of the Houston Longitudinal Congenital CMV Study as well as their physicians and the members of the Congenital CMV Longitudinal CMV Study Group for their support and dedication to that study which was begun in 1982. The Congenital CMV Longitudinal Study Group through the years has included: Shahzad Ahmed, Hanna Baer, MD, Amit R. Bhatt, MD, Peggy Blum, AuD and Texas Children's Hospital Audiology, Frank Brown, MD, Francis Catlin, MD, Alison C. Caviness, MD, PhD, MPH, David K. Coats, MD, Jane C. Edmonds, MD, Marily Flores, MS, Daniel Franklin, MD, Cindy Gandaria, Jewel Greer, Carol Griesser, RN, Mohamed A. Hussein, MD, Isabella Iovino, PhD, Allison Istas, MPH, Haoxing (Douglas) Jin, Mary K. Kelinske, OD, Joseph T. Klingen, Antone Laurente, PhD, Thomas Littman, PhD, Mary Murphy, MS, Jerry Miller, PhD, Christopher Nelson, MD, Daniel Noyola, MD, Evelyn A. Paysse, MD, Alan Percy, MD, Sara Reis, RN, Ann Reynolds, MD, Judith Rozelle, MS, O'Brien Smith, PhD, Paul Steinkuller, MD, Marie Turcich, MS, Sherry Sellers Vinson, MD, Robert G. Voigt, MD, Bethann Walmus, Jill Williams, MA, Daniel Williamson, MD, Kimberly G. Yen, MD, Martha D. Yow, MD, and Gail J. Demmler-Harrison MD.

## Author Contributions

**Conceptualization:** Ryan Rochat, Elizabeth Goodman, Jerry Miller, Wei Wang, Gail J. Demmler-Harrison.

**Data curation:** Ryan Rochat, Jerry Miller, Gail J. Demmler-Harrison.

**Formal analysis:** Ryan Rochat, Jerry Miller, Gail J. Demmler-Harrison.

**Funding acquisition:** Ryan Rochat, Jerry Miller, Gail J. Demmler-Harrison.

**Investigation:** Ryan Rochat, Elizabeth Goodman, Jerry Miller, Gail J. Demmler-Harrison.

**Methodology:** Ryan Rochat, Elizabeth Goodman, Jerry Miller, Gail J. Demmler-Harrison.

**Project administration:** Elizabeth Goodman, Wei Wang.

**Resources:** Ryan Rochat, Jerry Miller, Gail J. Demmler-Harrison.

**Supervision:** Elizabeth Goodman, Wei Wang, Gail J. Demmler-Harrison.

**Validation:** Ryan Rochat, Jerry Miller, Gail J. Demmler-Harrison.

**Visualization:** Ryan Rochat, Jerry Miller, Gail J. Demmler-Harrison.

**Writing – original draft:** Ryan Rochat, Elizabeth Goodman, Jerry Miller, Wei Wang, Gail J. Demmler-Harrison.

**Writing – review & editing:** Ryan Rochat, Elizabeth Goodman, Jerry Miller, Wei Wang, Gail J. Demmler-Harrison.

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
