## [Decision Letter · Decision Letter 0]

17 Sep 2024

PONE-D-24-25337­­­Health Outcomes and Healthcare Resource Use Related to ­­­Special Education Needs in Congenital CMVPLOS ONE

Dear Dr. Rochat,

Thank you for submitting your manuscript to PLOS ONE. After careful consideration, we feel that it has merit but does not fully meet PLOS ONE’s publication criteria as it currently stands. Therefore, we invite you to submit a revised version of the manuscript that addresses the points raised during the review process.

We look forward to receiving your revised manuscript.

Kind regards,

Fumihiko Namba

Academic Editor

PLOS ONE

Journal Requirements:

4. Thank you for stating the following financial disclosure: “Funding for this research was provided by Merck Sharp & Dohme Corp., a subsidiary of Merck & Co., Inc., Rahway, NJ, U.S.A. Merck Sharp & Dohme Corp BCM-Contract VEAP 7973 Demmler-Harrison (PI) 12/2019-current Burden of disease associated with congenital CMV disease: Congenital CMV Cost of Illness Study”

5. Thank you for stating the following in the Competing Interests section: “Co-authors Drs. Goodman and Wang are employees of Merck Sharp & Dohme Corp., a subsidiary of Merck & Co., Inc., Rahway, NJ, USA and may hold stock in Merck & Co., Inc., Rahway, NJ, USA. Drs. Rochat, Miller, and Demmler-Harrison received research support from Merck & Co, Inc. Dr. Demmler-Harrison has also received research support from Microgen Laboratories.”

We note that one or more of the authors are employed by a commercial company: Merck Sharp & Dohme Corp

a. Please provide an amended Funding Statement declaring this commercial affiliation, as well as a statement regarding the Role of Funders in your study. If the funding organization did not play a role in the study design, data collection and analysis, decision to publish, or preparation of the manuscript and only provided financial support in the form of authors' salaries and/or research materials, please review your statements relating to the author contributions, and ensure you have specifically and accurately indicated the role(s) that these authors had in your study. You can update author roles in the Author Contributions section of the online submission form. Please also include the following statement within your amended Funding Statement. “The funder provided support in the form of salaries for authors [insert relevant initials], but did not have any additional role in the study design, data collection and analysis, decision to publish, or preparation of the manuscript. The specific roles of these authors are articulated in the ‘author contributions’ section.” If your commercial affiliation did play a role in your study, please state and explain this role within your updated Funding Statement.

b. Please also provide an updated Competing Interests Statement declaring this commercial affiliation along with any other relevant declarations relating to employment, consultancy, patents, products in development, or marketed products, etc. Within your Competing Interests Statement, please confirm that this commercial affiliation does not alter your adherence to all PLOS ONE policies on sharing data and materials by including the following statement: "This does not alter our adherence to PLOS ONE policies on sharing data and materials.” (as detailed online in our guide for authors http://journals.plos.org/plosone/s/competing-interests) . If this adherence statement is not accurate and there are restrictions on sharing of data and/or materials, please state these. Please note that we cannot proceed with consideration of your article until this information has been declared. Please include both an updated Funding Statement and Competing Interests Statement in your cover letter. We will change the online submission form on your behalf.

6. We note that you have indicated that there are restrictions to data sharing for this study. PLOS only allows data to be available upon request if there are legal or ethical restrictions on sharing data publicly. For more information on unacceptable data access restrictions, please see http://journals.plos.org/plosone/s/data-availability#loc-unacceptable-data-access-restrictions. Before we proceed with your manuscript, please address the following prompts: a) If there are ethical or legal restrictions on sharing a de-identified data set, please explain them in detail (e.g., data contain potentially identifying or sensitive patient information, data are owned by a third-party organization, etc.) and who has imposed them (e.g., a Research Ethics Committee or Institutional Review Board, etc.). Please also provide contact information for a data access committee, ethics committee, or other institutional body to which data requests may be sent. b) If there are no restrictions, please upload the minimal anonymized data set necessary to replicate your study findings to a stable, public repository and provide us with the relevant URLs, DOIs, or accession numbers. For a list of recommended repositories, please see https://journals.plos.org/plosone/s/recommended-repositories. You also have the option of uploading the data as Supporting Information files, but we would recommend depositing data directly to a data repository if possible. We will update your Data Availability statement on your behalf to reflect the information you provide.

Reviewers' comments:

Reviewer's Responses to Questions

**Comments to the Author**

1. Is the manuscript technically sound, and do the data support the conclusions?

Reviewer #1: Yes

Reviewer #2: Yes

2. Has the statistical analysis been performed appropriately and rigorously? 

Reviewer #1: Yes

Reviewer #2: Yes

3. Have the authors made all data underlying the findings in their manuscript fully available?

Reviewer #1: No

Reviewer #2: Yes

4. Is the manuscript presented in an intelligible fashion and written in standard English?

Reviewer #1: Yes

Reviewer #2: Yes

5. Review Comments to the Author

Reviewer #1: This is an important paper demonstrating the global impact of cCMV whether or not asymptomatic at birth on the long term outcomes of children. The breakdown of specific conditions that qualify for IDEA is interesting and helpful to explore the variety of needs these children can have. There are a few clarifying questions that arise from the paper.

1. How did you identify the AcCMV cohort in the clinical arm since they were not part of the large prospective Houston CMV study?

2. Were ICD9 and ICD 10 codes used or were the ICD9 codes from the historical cohort reclassified into ICD10? It seems you mainly used the ICD10 codes to translate to the IDEA qualifying conditions.

3. Why do you suppose there was a much higher percent in the AcCMV who met IDEA criteria in the clinical cohort? Could that be related to the exclusion criteria mentioned in the historical group?

the tables are helpful for clarity and the study limitations are well addressed.

Reviewer #2: This study tried to find need for special education in children suffering from congenital cytomegalovirus infection, a nice study explored underaddressed issues. hence I have few observation.

1. Title need to be self explanatory for better understanding of readers

2.Limitation of study need to be brief.

Others seems ok

6. PLOS authors have the option to publish the peer review history of their article (what does this mean?). If published, this will include your full peer review and any attached files.

Reviewer #1: No

Reviewer #2: No

---

## [Author Response · Author response to Decision Letter 0]

11 Oct 2024

Response to Reviewers

Reviewer 1:

This is an important paper demonstrating the global impact of cCMV whether or not asymptomatic at birth on the long term outcomes of children. The breakdown of specific conditions that qualify for IDEA is interesting and helpful to explore the variety of needs these children can have. There are a few clarifying questions that arise from the paper. 

Thank you for these comments. We are glad to see that the reviewer agrees with an approach looking at the real-world utilization of these children as a marker of the needs of these children, specifically those classified as asymptomatic at birth. We hope our responses to your questions below are of help and we will make sure that they are addressed within the manuscript as well.

1. How did you identify the AcCMV cohort in the clinical arm since they were not part of the large prospective Houston CMV study?

Children with AcCMV in the clinical cohort were classified, similarly to those in the historical cohort, as those who did not have any of the following (lines 153-157): purpura/petechiae, jaundice, hepatosplenomegaly, microcephaly, unexplained neurological abnormality, elevated liver enzymes, hyperbilirubinema, hemolytic anemia or thrombocytopenia. Since we were not routinely screening for CMV in the clinical cohort, these were patients who came into our care through referral for any of a variety of reasons, either failed hearing screening at birth, or screening through parental concern. We have amended the manuscript in the methods section to highlight this fact.

2. Were ICD9 and ICD 10 codes used or were the ICD9 codes from the historical cohort reclassified into ICD10? It seems you mainly used the ICD10 codes to translate to the IDEA qualifying conditions.

For the historical cohort, the codes we used were manually abstracted from historical records (typically upon review of paper charts) into the ICD10 ontology. As such, there was no need to translate from ICD9 to ICD10. For the clinical cohort, all codes were extracted from the electronic medical record. In accordance with regulations from the Centers for Medicare & Medicaid Services, all codes within our system were migrated from ICD9 to ICD10 years prior to our extraction. In our system, this migration was retrospective, and so when extracting these codes in 2021, the ICD codes had already been translated into the ICD10 ontology. We have modified the text of the manuscript in the methods section to clarify that all codes were ICD10 when manually abstracted at time of extraction.

3. Why do you suppose there was a much higher percent in the AcCMV who met IDEA criteria in the clinical cohort? Could that be related to the exclusion criteria mentioned in the historical group? 

The classification of AcCMV and ScCMV was similar across the two cohorts, with AcCMV being those infants with positive CMV testing who did not have any of the following at birth: purpura/petechiae, jaundice, hepatosplenomegaly, microcephaly, unexplained neurological abnormality, elevated liver enzymes, hyperbilirubinema, hemolytic anemia or thrombocytopenia. Since AcCMV cases in the historical cohort were identified in through universal screening, it is possible that this approach was more sensitive in identifying patients than what was seen in the clinical cohort which represents a real-world view of referrals for this congenital infection. As we note in the limitations section, the clinical cohort is a reflection of referrals to center, and so there is the potential for referral bias in this group. This being said, we feel that the higher percentage in the AcCMV who met criteria in the clinical cohort is due to the real-world evaluation of these children. What that means is that in comparing the historical to clinical cohort, in the historical cohort these codes would have been abstracted upon review of documentation from that study, where as the codes for the clinical cohort came from tens of thousands of provider encounters across dozens of specialties. This is not to say that the documentation in the historical cohort was not as complete, merely that the historical documentation was from a single provider throughout the course of that study, versus a reflection of the evaluations of thousands of providers in the contemporary clinical cohort. We have added a section in the discussion to address this.

Reviewer 2:

This study tried to find need for special education in children suffering from congenital cytomegalovirus infection, a nice study explored underaddressed issues. hence I have few observation.

Thank you for these comments. We hope our responses to your questions below are of help and thank you for helping improve the content and readability of the paper.

1. Title need to be self explanatory for better understanding of readers

We have changed the title to make it more self explanatory. The current title is now: 

Healthcare Outcomes and Special Education Eligibility in Children with Congenital CMV

2.Limitation of study need to be brief. 

We have shortened this section as requested.

---

## [Editor Report · Decision Letter 1]

21 Oct 2024

Healthcare Outcomes and Special Education Eligibility in Children with Congenital CMV

PONE-D-24-25337R1

Dear Dr. Rochat,

We’re pleased to inform you that your manuscript has been judged scientifically suitable for publication and will be formally accepted for publication once it meets all outstanding technical requirements.

Kind regards,

Fumihiko Namba

Academic Editor

PLOS ONE
---

## [Editor Report · Acceptance letter]

29 Nov 2024

PONE-D-24-25337R1 

PLOS ONE

Dear Dr. Rochat, 

I'm pleased to inform you that your manuscript has been deemed suitable for publication in PLOS ONE. Congratulations! Your manuscript is now being handed over to our production team.

Kind regards, 

on behalf of

Dr. Fumihiko Namba 

Academic Editor

PLOS ONE